# Robustness Analysis of a Fast Virtual Temperature Sensor Using a Recurrent Neural Network Model Sensitivity

**DOI:** 10.3390/s25237193

**Published:** 2025-11-25

**Authors:** Patryk Chaber, Bartosz Chaber

**Affiliations:** 1Faculty of Electronics and Information Technology, Warsaw University of Technology, 00-665 Warsaw, Poland; 2Faculty of Electrical Engineering, Warsaw University of Technology, 00-662 Warsaw, Poland; bartosz.chaber@pw.edu.pl

**Keywords:** virtual sensor, recurrent neural network, automatic differentiation

## Abstract

Virtual sensing is an emerging field of research that has garnered increasing attention in recent years. In this paper, we focus our attention on recurrent neural networks for time-series forecasting, namely, the Nonlinear AutoRegressive eXogenous model (NARX). The NARX is utilized as a surrogate neural network for simulating heat flow. Our research has investigated the sensitivity of NARX models of varying complexity. The presented results show that the loss function value alone does not indicate the model’s sensitivity. We have demonstrated that undertrained models exhibit visible artifacts in their sensitivity, highlighting the model’s weak points. From observing how the sensitivity changes over training epochs, we can conclude that the sensitivity increases with more epochs, while its overall shape remains relatively unchanged.

## 1. Introduction

Extensive monitoring of a physical process is often highly beneficial; however, it requires considerable effort due to the high number of sensors installed and maintained in various measurement locations. Although a vast network of sensors enables the rapid detection of anomalous events, the high cost and limited lifetime of these sensors often make implementation impractical. Using virtual sensors instead of real ones alleviates the cost and practicality issues, allowing virtual measurements to be made without affecting the process.

Virtual sensing is carried out within a computer simulation that matches the state of a physical process. As such, the simulations are often time-consuming, and much attention has been focused on using faster surrogate models, such as recurrent neural networks (RNNs), for time-series forecasting. Given that the recurrent neural network model is differentiable, we can calculate its sensitivity to the model’s input signals using automatic differentiation algorithms. The high sensitivity of the model might show weak points of the model, where it is more prone to raise a false anomaly alert as the response to a slight control signal change might be exaggerated. However, in general, we will call a model robust if its sensitivity does not exhibit behaviour that can’t be explained by the dynamics of the process being modeled.

In our research, we are focusing on a recurrent neural network model, in essence, the Nonlinear AutoRegressive eXogenous model (NARX) architecture, and how its structure influences its sensitivity. We have chosen NARX as it has proven to correctly match the dynamics of the gas flow in the pneumatic system. Additionally, it has a relatively simple architecture, thus requiring lower computational resources compared to other RNN architectures. Moreover, due to its straightforward structure, sensitivity analysis is easier to perform compared to models such as LSTM. We examine the influence of the model complexity and the number of training epochs on the sensitivity values for a given input signal. In this paper, we propose a new tool that enables the comparison of models’ responses to perturbations of input signals under various conditions. Based on the sensitivity plots and a general knowledge of the modeled process dynamics, one can pinpoint where the model poorly mimics the original process’s behaviour.

The paper is structured as follows: a literature review of different approaches to implement virtual sensors is presented in Section 2. The physical process is then described in Section 3, while its thermohydraulic model is presented in Section 4. After that, Section 5 shows how data from the previous section are used for training our RNN model, and how we calculate the model’s sensitivity. In Section 6, we explain the experiments carried out to observe the effect of the RNN model’s complexity on its sensitivity. We discuss the obtained results in Section 7, and conclude the findings in Section 8, where we also note future plans.

## 2. Literature Overview

As a result of an ongoing digital transformation, virtual sensing has garnered increasing attention over the last decade. We could classify the approaches based on the underlying foundation of the sensor’s implementation into two classes: model-based sensors and data-driven sensors [1,2].

In the case of model-based sensors, the virtual measurement is evaluated within a physical simulation. As the virtual sensor should provide data in real-time, the discretization of the model must be adjusted [3], or a reduced-order model [2,4] may be used to speed up calculations at the cost of slightly lower fidelity.

The other way of improving virtual sensor performance is using data-driven models, where the mapping between measurements and target variables is learned directly from data. Instead of solving complex equations to obtain soft sensor measurements, this model utilizes statistical and machine learning techniques to capture the process dynamics. As this approach relies on the quality of training data, if it is lacking, the model exhibits limited robustness and poor output accuracy in the presence of unseen conditions. Data-driven models commonly utilize Recurrent Neural Networks (RNN) in the form of NARX [5], Long-Short Term Memory model (LSTM) [6], Gated Recurrent Unit model (GRU) [7], although Convolutional Neural Network models (CNN) are also quite often used [8,9]. However, soft sensors are not limited to artificial neural networks; approaches utilizing Support Vector Regression [10], Decision Trees [11], or Partial Least Squares [12] are also commonly employed for this purpose.

In the case of artificial neural network models, the effort required for training the model allows for much faster inference times compared to a regular physical simulation. The training dataset should be sufficiently large to enable efficient training of the neural network’s weights, thereby minimizing the loss function while achieving a reasonable level of generalization capability. In cases of long-lasting experiments on real devices, computer simulation may be the source of a training dataset. Otherwise, raw measurements of a physical process might be utilized to optimize the neural network’s weights.

It is worth emphasizing that both model-based and data-driven virtual sensors have a number of applications in industry and are often an integral part of industrial systems. Authors of [13,14] have implemented a virtual temperature sensor that utilizes a computational fluid dynamics real-time simulation for monitoring and controlling the environment inside a greenhouse. Contrary to the model-based virtual sensor approach, the authors in [1] based their virtual flow sensor on a recurrent neural network, rather than a physical model. The artificial neural network was trained on real measurement data, and the resulting model was then utilized for control purposes.

These sensors are used as a replacement for physical sensors that would be located in hard-to-reach places [15,16], and are used as redundancy, i.e., to verify the operation of the actual sensor [1,17,18]. Virtual sensors can also be used to estimate physical quantities based on other actual measurements, e.g., to enrich the model used in model predictive control [1,18,19]. Virtual sensors can also be used as a prototype or proof of concept during the design process for new physical sensors [20]. Model-based soft sensors are also helpful as a substitute for real measurements when gathering training data for data-driven virtual sensors. In [21,22,23], mathematical models have been used for generating a training dataset for an LSTM neural network acting like a soft sensor for chemical parameters in a wastewater treatment plant. Research presented in [24] describes a thermohydraulics simulation being used to generate a training dataset for a Deep Operator Network model of a hot leg of a Pressurized Water Reactor.

As the response to the soft sensing field becoming more complex and advanced, new tools for automatic machine learning are being developed [25,26], which enable further expansion of use cases for virtual sensors and underscore the necessity for careful and comprehensive verification of their quality.

The black-box nature of RNN-based virtual sensors has been mentioned in [27], where it has been integrated with a Kalman filter to form a safer, hybrid sensor. A Kalman filter has also been used in [28] where it has helped build a virtual load sensor based on a linearized, multi-body mathematical model. As noted by [29], virtual sensors alleviate the high costs associated with the installation and maintenance of real sensors. They have presented a Long Short-Term Memory (LSTM) model for virtual pressure and temperature sensors, trained on real measurement data.

In summary, we can see that recurrent neural networks can be successfully used in virtual sensors. Unlike simulation models, RNNs enable real-time operation, which is very important in systems with short time constants. At the same time, it is worth noting that since RNN models are black boxes, it would be helpful to have a tool that would allow us to verify the predictability of this model’s operation in various states, even transitional ones.

Lastly, we have analyzed a set of recent papers concerning virtual sensors in industrial applications, and gathered results in Table 1. Papers were compared in terms of the model type used in the soft sensor, the domain on which the paper focused, and the Key Performance Indicators (KPIs) considered in the paper. It is worth noting that the majority of soft sensors are data-driven. This trend seems reasonable, especially if real-time operation is a necessity for the authors, which is often the case in industrial applications. Application domains range from smart homes and IoT, through the automotive industry, to wastewater treatment facilities and hydraulic thermal plants. This vast range of uses confirms the wide applicability of the soft sensors in almost any area where measurements are gathered. Lastly, we have listed the KPIs that each paper focused on and highlighted those related to the robustness of the model, whether the model-based or data-driven approach was applied. As shown in the table, only a few papers addressed the problem of model generalization abilities and robustness, confirming the suspicion that this is not a main concern for researchers.

## 3. Physical Process Description

In our research, we have conducted experiments on a model of a simple ventilation system described in [43], which has been extended with a coupled thermal simulation. The mentioned ventilation system is a real physical laboratory stand, where a controllable fan drives a flow of air in a pipe system. A set of motorized valves allows controlling the flow rate in different branches of the system. The laboratory stand consists of two interconnected circuits: a cold circuit and a hot circuit. The gas flow between the circuits is controlled with the motorized valves. A controllable heating element can heat the hot circuit. The gas outflows through three output pipes, where the cold and hot air mix in different proportions. A diagram presenting the simulated heating and ventilation system is shown in Figure 1. The arrows present the inflow and outflow of the ambient air. The valve connected to the third output pipe has been set to a fully open position. The percentage rotation of the disc between the closed and open states in the remaining two butterfly valves (MOV-12 and MOV-22) can be controlled by an external signal. We have chosen only two valves for control to limit the complexity of our problem. On the other hand, we have decided to use $T_{3}$ as the temperature sensor to make its relationship to the valves’ positions more interesting.

Modeling of the heat flow is an interesting problem, as the process of heating has a much higher time constant than the gas flow alone. It is worth noting that changes in valve positions affect the flow rates in different branches of the pipe system, which in turn affects the speed of the heat flow. As [43] proved that recurrent neural networks can learn how flow changes based on the valve’s positions, we aim to further this work by predicting heat flow as well. Gathering measurement data for implementing our virtual temperature sensor, for the presented physical process, requires dealing with two main challenges: conducting long-lasting experiments to gather sufficient measurement samples, and faithfully recreating the crucial components of the real pipe system to produce an accurate numerical model. For these two reasons, this work focuses solely on building a computer simulation of the heating and ventilation system. We have aimed to recover the general dynamics of the system, as presented in Figure 2. The figure shows two different experimental scenarios with random valve positions and constant heater and fan powers. As we weren’t yet building an accurate numerical model, we had only compared the general dynamics of the real system. It can be seen that the simulation reproduces a smooth increase in temperature to the steady state.

## 4. Gas and Heat Flow Simulation

The physical model has been implemented in the Thermal Hydraulics Module, a component of the MOOSE Framework [44], which utilizes the Finite Volume Method (FVM) discretization of the Euler equations for gas flow in a system of interconnected pipes with junctions. Each pipe has been discretized with one-dimensional segments of Δl=10mm to balance computation time with result accuracy. The total number of cells in the FVM model is approximately 1500, which enables faster-than-real-time simulation. The open ends of output pipes have been prescribed an Outlet Boundary Condition that fixes the gas pressure value *p* = 101,325 Pa. The open end near the fan prescribes an Inlet Boundary Condition which fixes the gas temperature at *T* = 28.5 °C and the initial gas velocity as 0 m/s. Air is modeled as an ideal gas with molar mass 28.965197004 × 10^−3^ kg/mol and γ=1.4.

The governing equations solved by the Thermal Hydraulics Module are the Euler equations of a compressible fluid (or a gas) flowing inside a circular pipe of cross-sectional area *A*. The equations for conservation of mass, momentum, and total energy are as follows:$$\begin{array}{cc} \frac{\partial \rho A}{\partial t}+\frac{\partial \rho u_{l}A}{\partial l} & =0,\\ \frac{\partial \rho u_{l}A}{\partial t}+\frac{\partial \rho u_{l}^{2}A}{\partial l}+\frac{\partial pA}{\partial l} & =p\frac{\partial A}{\partial l}-FA+\rho g_{l}A,\\ \frac{\partial \rho EA}{\partial t}+\frac{\partial \rho u_{l}EA}{\partial l}+\frac{\partial u_{l}pA}{\partial l} & =pu_{l}g_{l}A, \end{array}$$
where *t* is time, *l* is the spatial position along the pipe axis, ρ is the gas density, $u_{l}$ is the velocity along the pipe axis, *E* is the specific total energy, *p* is the gas pressure, *F* is the viscous drag force density, and $g_{l}$ is the component of the acceleration due to gravity in the axial direction of the pipe element. Gravity acceleration vector is defined in the Cartesian coordinate system as (0, 0, −9.81 m/s^2^). In our simulation, the diameter of the pipes is constant along all pipes and equals *D* = 100 mm.

The flow equations are coupled with a heat transfer equation. The heat transfer is assumed to happen through a pipe’s wall heat flux $q_{\mathrm{wall}}$:(1)$$\begin{array}{c} q_{\mathrm{wall}}=H\left(T_{s}-T\right), \end{array}$$
where H is the heat transfer coefficient (calculated using Dittus-Boelter equation), $T_{s}$ is the heater’s surface temperature, and *T* is the air temperature. Our simulation utilizes a heater element that provides thermal energy to the system. For the purpose of testing the sensitivity of the virtual sensor to the valve positions, we have fixed the power of the heating element. Our simulation also has a fixed power of the fan driving the flow in the pipe system. Both the heater power and the fan power are linearly increased to their maximum values in the first 500 s of the simulation. This transient state is skipped for the training dataset, allowing the fan power to be assumed constant for the remainder of the research.

## 5. Model Design

For this research, we have focused our attention on neural networks of a recurrent nature, namely NARX. The model that was used had a structure visible in Figure 3.

This network consists of three layers: input, hidden, and output. The input layer consists of two parts (in the figure, colored with different hues). The first one is connected to exogenous inputs of the model (blue hue in the figure), while the second one is connected to the feedback signal from the model’s output (red hue in the figure). It is essential to notice that in both cases, what actually reaches the input layer is a series of delayed signals, not only the last value for each signal. The hidden layer of this network consists of a number of nodes, each utilizing a sigmoid activation function:(2)$$\varphi \left(v\right)=\frac{2}{\left(1+\exp(-2v)\right)}-1.$$

For the sake of better training, the input layer is preceded by a normalization in the form of min-max mapping, based on the minimum and maximum input values from the training dataset. Additionally, the target values from the training dataset were normalized; therefore, it was necessary to add a reverse min-max mapping function after the output layer.

Because of the discrete nature of NARX models, the following formulae utilize the discrete time variable *k*, instead of *t*, to underline this feature. The relation between those two variables is $t=t_{0}+kT_{\mathrm{sampling}}$, where $T_{\mathrm{sampling}}$ represents the sampling period, and $t_{0}$ denotes the initial time, which is often the same as the time of the experiment’s beginning. Assuming one output signal y(k), two input signals ($\vec{u}\left(k\right)=\left[u_{1}\left(k\right),u_{2}\left(k\right)\right]$, as $n_{u}=2$), and a general case of $n_{a}$ feedback dynamics order and $n_{b}$ input dynamics order, the model can be expressed as follows:(3)$$\begin{array}{cc} y^{F}\left(k\right) & =W_{H}\varphi \left(W_{X}\mu_{X}\left(x(k)\right)+W_{F}f\left(k\right)+b_{1}\right)+b_{2}, \end{array}$$(4)$$\begin{array}{cc} y(k) & =\mu_{Y}\left(y^{F}\left(k\right)\right),\; \end{array}$$
where input vector x(k) and feedback vector f(k) are defined as:(5)$$\begin{array}{cc} x(k) & ={\left[\vec{u}\left(k-1\right),\ldots ,\vec{u}\left(k-n_{b}\right)\right]}^{T},\; \end{array}$$(6)$$\begin{array}{cc} f(k) & ={\left[y^{F}\left(k-1\right),\ldots ,y^{F}\left(k-n_{a}\right)\right]}^{T}. \end{array}$$

Weights between layers are represented as follows:(7)WX=wX11…wX12nb⋮⋱⋮wXnh1…wXnh2nb,WF=wF11…wF1na⋮⋱⋮wFnh1…wFnhna,WH=wH1⋮wHnhT,
where wsji denotes the weight between input signal *i* from previous layer, and signal *j* of the next layer. Descriptor *s* tells which input signal the weights multiply i.e., X – x(k), F – f(k), and H—the output of hidden layer with $n_{h}$ neurons. Bias signals are denoted as vector $b_{1}={\left[b_{1}^{1},\ldots ,b_{1}^{n_{h}}\right]}^{T}$ and scalar $b_{2}$. Vector $\vec{w}$ contains all weights from matrices $W_{X}$, $W_{F}$, $W_{H}$, vector $b_{1}$ and scalar $b_{2}$.

As a training data set, a set of signals, MOV-12 and MOV-22, is used as input information ($u_{1}\left(t\right)$ and $u_{2}\left(t\right)$, respectively), and $T_{3}$ is used as a target output signal for the model (y(t)). All models were trained using Levenberg-Marquardt backpropagation (from MATLAB Deep Learning Toolbox v14.4) in multiple runs of 100 epochs. The only stopping criterion was reaching a set number of epochs, as our aim was not to achieve only properly trained models, but also those overtrained and undertrained. The training was performed using Intel^®^ Core™ i7-10700F CPU @ 2.90 GHz with 32 GB RAM. The data was split into two datasets – the first 85% of all samples were used to train, whereas the last 15% were utilized as a verification dataset. All models were trained in parallel on various machines, which resulted in different maximum epochs for each model. Due to limited resources, we were unable to train those models until the verification error began to rise.

The training dataset should cover all representative samples of a state space to enable a proper generalization of a trained neural network. Building such a dataset is time-consuming, which is why we have generated the data with the help of the FVM simulation. To prepare such a vast dataset, we ran numerous simulations with random control signals on separate computing machines and then stitched the results into a single dataset. After each control change, there was sufficient time for the signal to stabilize (approximately 500 s), allowing the model to learn the static characteristics of the process. Also, the data were long enough to cover as much of the input signals’ range as possible – in total, the whole dataset consisted of almost 273 thousand samples, collected at a frequency of 1 Hz. Distribution of input signals for the training dataset is shown in Figure 4. In general, control signal values were generated as random steps from a uniform distribution; however, an increased number of samples for control signals $u_{1}=u_{2}=0.95$ can be observed in the histograms. To allow seamless stitching, each dataset started and ended at the same operating point, resulting from the aforementioned control signal values, which, in turn, visibly influenced the histogram, as shown in the figure.

### Calculating Gradient of the Recurrent Neural Network

To calculate the sensitivity of the RNN, we calculate the gradient of the output of the model (temperature $T_{3}\left(t\right)$, being a scalar) with respect to the two input control signals $u_{1}\left(t\right)$ and $u_{2}\left(t\right)$, using automatic differentiation. This way, we determine the influence of a change in input signals on the output value. Control signals may sometimes be affected by disturbances, so information about their sensitivity is essential. Automatic differentiation is a set of techniques that apply the chain rule for calculating derivatives of composite functions, which also applies to neural networks. In most cases, the gradient of a scalar loss function is calculated with respect to the network’s weights. In this case, the so-called reverse mode automatic differentiation needs only a single evaluation of the loss function to obtain all elements of its gradient with respect to the weights. On the other hand, a forward mode automatic differentiation (in general) requires as many evaluations as there are weights of the network. Most of the forward-mode automatic differentiation libraries are implemented based on a new datatype called dual numbers [45] or its extension that allows tracking multiple derivatives with a single function evaluation, i.e., hyper-dual numbers [46]. In our research, we use the latter for tracking two perturbations $\vec{\epsilon }=\left(\epsilon_{1},\epsilon_{2}\right)$ for each of the two control signals related to the respective valves.

The process of calculating a derivative starts with seeding (seeding means initializing a dual number with some perturbation, to see how it propagates through the differentiated function) of the control signal sample $\vec{u}\left(t\right)+\vec{\epsilon }=\left(u_{1}\left(t\right)+\epsilon_{1},u_{2}\left(t\right)+\epsilon_{2}\right)$. In contrast to regular neural networks, recurrent neural networks return both an output after a single step, y(k), and a delayed output signal that becomes a feedback input to the network’s feedback input layer, resulting in a memory of the model. After evaluation of $y^{F}\left(k\right)=g\left(\vec{w},\vec{u}\left(k\right),\vec{u}\left(k-1\right),\ldots ,\vec{u}\left(k-n_{b}\right),y^{F}\left(k-1\right),\ldots ,y^{F}\left(k-n_{a}\right)\right)$, where $\vec{w}=\left(W_{H},W_{X},W_{F},b_{1},b_{2}\right)$, the derivatives $\frac{\partial y^{F}\left(t\right)}{\partial \vec{u}\left(t\right)}$ are readily available. It must be mentioned that by evaluating the recurrent neural network further, we will be provided with $\frac{\partial y^{F}\left(t+\tau \right)}{\partial \vec{u}\left(t\right)}$, which we utilize as a measure of sensitivity.

In our work, we seed every 100th control signal sample. By repeating the procedure 100 times, and shifting the starting sample number each time, we can calculate all $\frac{\partial y(t+\tau )}{\partial \vec{u}\left(t\right)}$ for every τ∈0,1,…,100. The number of the maximum value of τ is set arbitrarily, and has to balance two important factors: time being spent computing sensitivities of the model, and the number of seconds after which we neglect the effect of a control signal $\vec{u}\left(t-\tau \right)$. If we choose a too small value of τ, we might expect a kind of “aliasing” of a non-decayed influence of some past control signals. In our work, we have chosen to use a seeding interval of 100, as for most of the trained models, the sensitivity after τ=100 has been approximately an order of magnitude smaller than its strongest value.

## 6. Experiments

In our research, we have examined the impact of several factors on the overall sensitivity of the model, as defined in the previous sections. The scope was limited to the following properties of the NARX model:Number of hidden neurons ($n_{h}$),Dynamics order ($n_{a}=n_{b}$),Number of epochs spent training the network.

Sensitivity metric was determined for each configuration, for each input signal, and for a range of delays (τ). In this section, a subset of the most prominent results is presented. For the sake of clarity, only selected results are presented, specifically, most results focus on the dynamics order of $n_{a}=n_{b}=4$, as there was no straightforward relation between this parameter and the sensitivity of the model.

First, it is worthwhile to examine a plot of the sensitivity signal in the context of the verification data. In the Figure 5, there are input signals of the model (bottom), a reference and an output signal of the model (second from the bottom), and partial derivatives of the output signal of the model with respect to the two input signals, at different time delays. This data is for a model with $n_{a}=n_{b}=4$, $n_{h}=10$, evaluated after 1500 epochs of training, although this specific model serves as just a general example. In a close-up of the sensitivity values and the control signal (Figure 5), it is visible that the value of sensitivity peaks near the transient states of the model, specifically when the input signals change the most. Nevertheless, there is no clear relationship between the magnitude of the change in the input signal and the values of the derivatives.

It has to be noted that this variation in values of sensitivity changes over the course of models’ training, as is shown in Figure 6. In the figure, the delay parameter was τ=3, as this is the value for which the sensitivity was the largest. There are visible changes in sensitivity across a range of epochs, both in terms of steady-state values and transient states.

Although the most notable behaviors were observed during the model’s transient states, there were also structures where the sensitivity value changed rapidly over an extended time period. This was mostly visible for undersized models, e.g., $n_{a}=n_{b}=4$, $n_{h}=5$, as visible in Figure 7. This observation, therefore, helps differentiate two types of sensitivity problems: one where the value of this metric is large but corresponds to a short time period, and the second, where this state of higher sensitivity is extended over a larger time period. In our research, the latter is more common with undersized and undertrained models, whereas more complex and better-trained models tend to exhibit short but pronounced sensitivity spikes.

It is worthwhile to see the Mean Squared Error (MSE) of the model for verification data over the course of training, shown in Figure 8. None of the presented models have reached the stage of permanently losing their generalization capabilities, even though they were trained over an extended period of time, with an aim to overfit them. It is also the reason for training on the data directly from the simulations, without any additional noise.

We have compared the MSE values with the maximum values of sensitivity with respect to both delay and time for consecutive epochs of training; the results are displayed in Figure 9. It can be observed that as the value of MSE decreases, the maximum value of sensitivity increases for both input signals of the model. Although the relation is not linear, the overall trend seems to be consistent.

If we look at the maximum values of the partial derivative with respect to time, and show them in relation to delays and training epochs (see Figure 10), we can see that the shape (along the τ axis) of the derivative is consistent after some initial training epochs. The only variation is in the height of consecutive plots. One can observe that the maximum value of the sensitivity is always for the highest epoch number and for τ=3, which corresponds to the peak of the sensitivity for the network. The complexity of the neural model, i.e., the number of hidden neurons, did not influence the shape or values of the sensitivity in any visible way.

During this research, we have also explored the influence of dynamic orders on the neural network model’s sensitivity, specifically testing $n_{a}=n_{b}=2$ and $n_{a}=n_{b}=8$. Models with $n_{a}=n_{b}=2$ have not manifested more extended periods of time of rapidly changing sensitivity that have been seen for undertrained models with $n_{a}=n_{b}=4$. Models with $n_{a}=n_{b}=8$ have also been susceptible to such extended time periods of significant sensitivity. Otherwise, all the observations that were presented earlier for $n_{a}=n_{b}=4$ have been repeated for those structures as well.

## 7. Discussion

The conducted experiments reveal that the sensitivity of the NARX-based virtual sensor evolves significantly during training. Although the Mean Squared Error consistently decreases, high sensitivity regions might be present in places that are not justified by the process dynamics, demonstrating that a lower loss value does not necessarily correspond to a robust model. This lack of relationships indicates that conventional loss-based evaluation is insufficient for assessing the reliability of recurrent neural network (RNN) models used in control-oriented virtual sensing.

Sensitivity peaks occur predominantly during transient states, reflecting the model’s response to rapid changes in input. These regions correspond to areas of higher model uncertainty and should be considered potential weak points in prediction confidence. Two distinct behaviors were identified: short, high-magnitude sensitivity spikes, which typically appear in well-trained models capturing dynamic transitions, and extended periods of elevated sensitivity, characteristic of undertrained or undersized architectures.

To summarize the findings, we present in Figure 11a NARX model ($n_{a}=n_{b}=4$, $n_{h}=5$) that exhibits four distinct characteristics during training. In the figure, we plot a change of sensitivities over time, where (for the given virtual temperature sensor in a pipe system) we expect high sensitivity changes at times when we switch control signals. The sensitivities are calculated only for τ=0, thereby limiting our analysis to the immediate effect of the control change. To make the plot easier to interpret, we focus only on the sensitivity for the $u_{1}$, as the other control signal gives similar results.

At the beginning of the training, after only 300 epochs, Figure 11a presents that the model lacks the proper sensitivity at control switching time instants. Moreover, a growing sensitivity change before *t* = 3500 s is an indication of low robustness, as the change does not correspond to the physics of the process. Figure 11b shows that after 700 training epochs, the sensitivity to control switching is properly recovered; however, the sensitivity change grows even faster between *t* = 3000 s and *t* = 3500 s, which again signals that such a model is not robust. The NARX model is robust, as can be seen in Figure 11c, after training it for 1500 epochs. The sensitivity is constant between the control signal changes every 500 s. Figure 11d shows that the model, after 7600 epochs, remains robust, and its sensitivity continues to adjust during the training process.

Exploiting the low robustness of the presented models in Figure 11a,b is currently outside the scope of our research. Still, it should be theoretically possible to craft a malicious control signal that exploits the higher sensitivity not grounded in the modelled process dynamics.

Overall, the analysis confirms that sensitivity-based evaluation provides complementary insight into model robustness. It enables the identification of undertrained or unstable regions that remain undetected by standard performance metrics. Incorporating sensitivity analysis into the training process may therefore enhance the reliability of neural virtual sensors intended for real-time industrial applications.

## 8. Conclusions

In this research, we have analyzed the robustness of a recurrent neural network-based virtual temperature sensor through sensitivity evaluation with respect to control inputs. Our experiments focused on a particular physical problem (heat and gas flow in a pipe system), using a single type of recurrent neural network model (Nonlinear AutoRegressive eXogenous model). The results highlight sensitivity analysis as a valuable complement to conventional performance metrics, enabling the identification of weakly trained or unstable regions within the model. Incorporating such analysis into training procedures may improve the prediction confidence of neural virtual sensors used in real-time control applications.

In the future, we plan to investigate how other types of recurrent cells (LSTM, GRU) affect the sensitivity, as the cells are often used to alleviate the problem of vanishing/exploding gradients, so we expect higher sensitivity for the architectures. Another direction for exploration is testing different optimization algorithms for training RNNs.

## Figures and Tables

**Figure 1 sensors-25-07193-f001:**
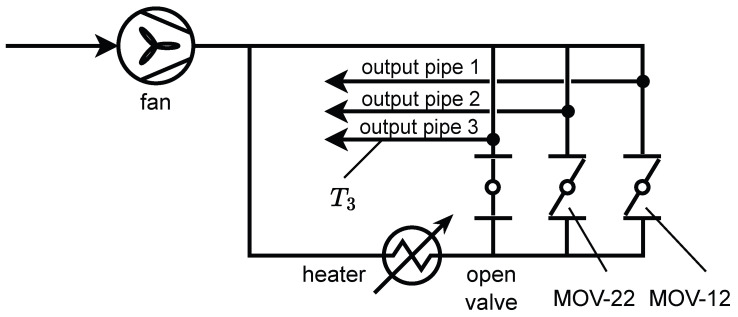
Pneumatics diagram of the simulated heating and ventilation system.

**Figure 2 sensors-25-07193-f002:**
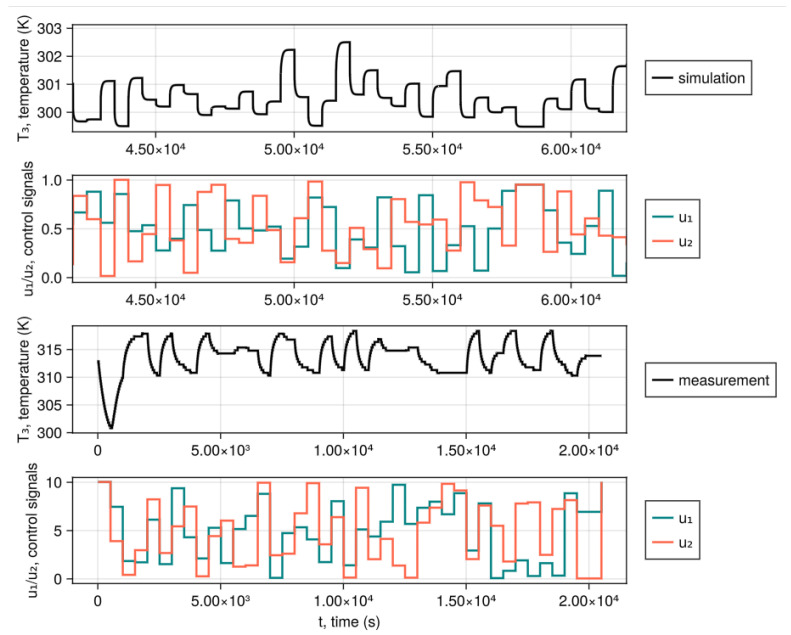
Dynamics comparison between a simulated heat flow and measurement data for random valve positions.

**Figure 3 sensors-25-07193-f003:**
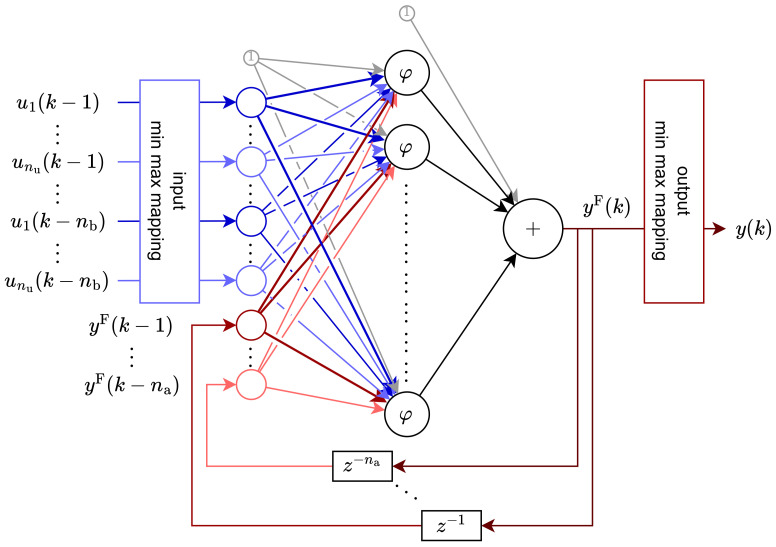
(**Color online**) General NARX structure with delayed inputs (blue hues), feedback (red hues), and bias (grey color) signals.

**Figure 4 sensors-25-07193-f004:**
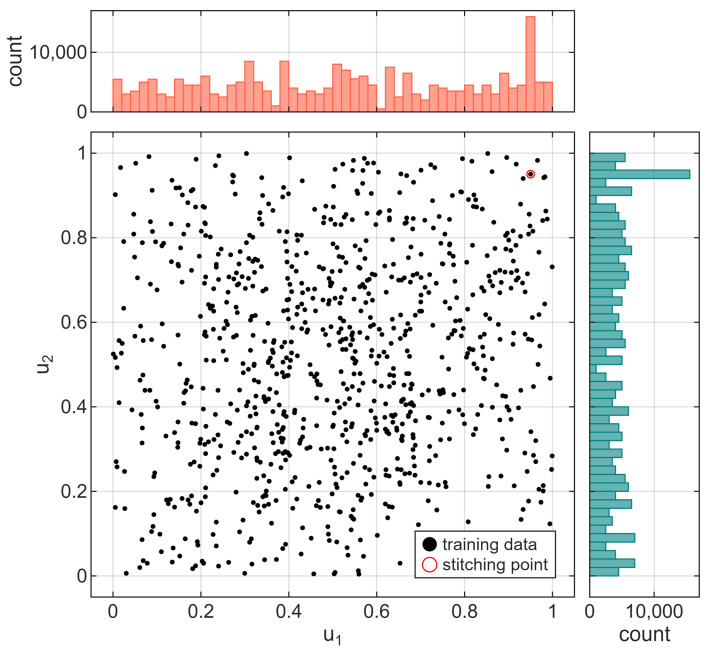
Distribution of input signal values for training dataset.

**Figure 5 sensors-25-07193-f005:**
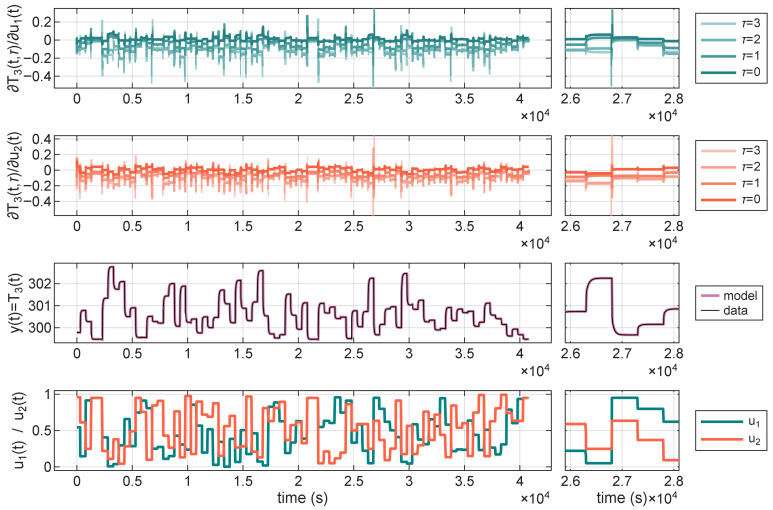
Derivatives of the output temperature $T_{3}$ (at different time delays τ) with respect to the two control signals $u_{1}$ (**first row**) and $u_{2}$ (**second row**), the output of the network (**third row**), and the control signals over time (**forth row**).

**Figure 6 sensors-25-07193-f006:**
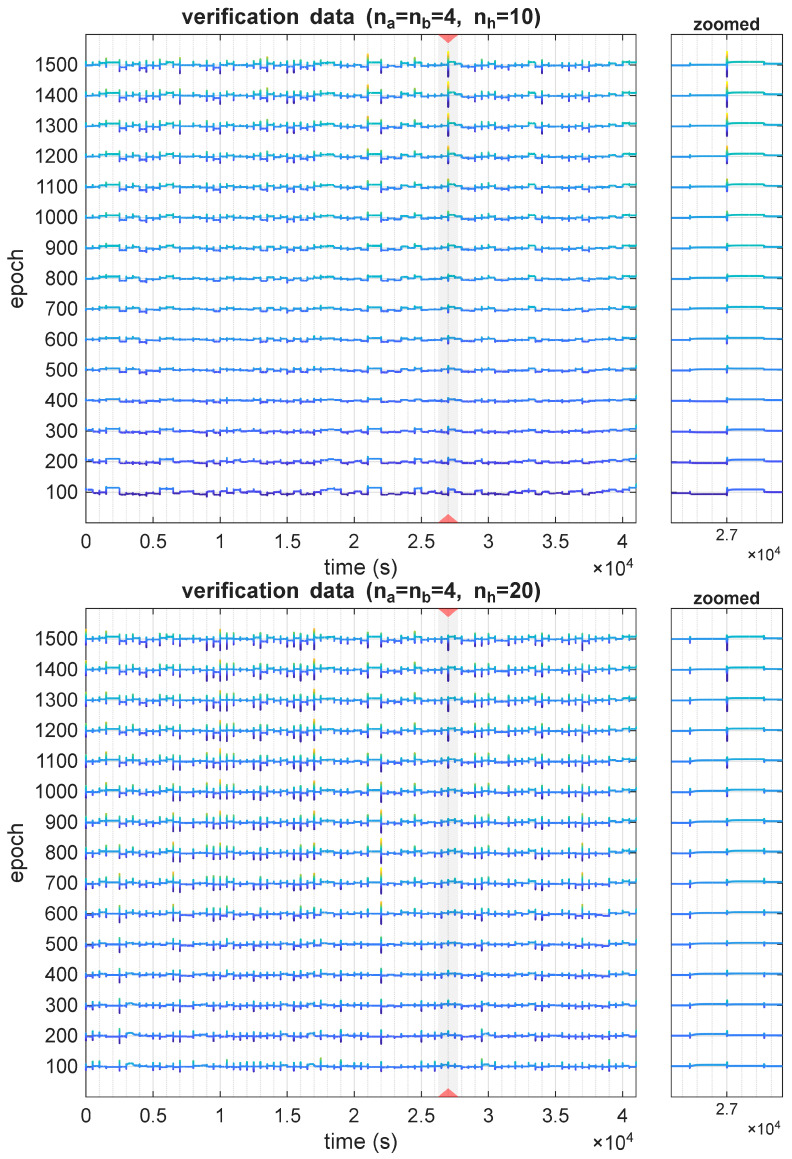
Derivatives $\partial T_{3}\left(t,\tau =3\right)/\partial u_{1}\left(t\right)$ for different time instants, plotted for different training epochs of the architectures with $n_{h}=10$ (**top**), and $n_{h}=20$ (**bottom**).

**Figure 7 sensors-25-07193-f007:**
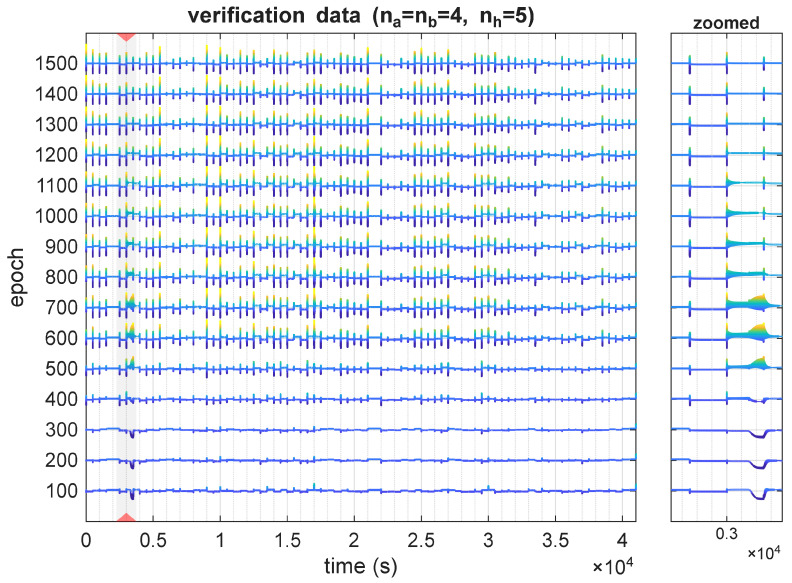
Derivative $\partial T_{3}\left(t,\tau =3\right)/\partial u_{1}\left(t\right)$ for different time instants, plotted for different training epochs of the architecture with $n_{h}=5$.

**Figure 8 sensors-25-07193-f008:**
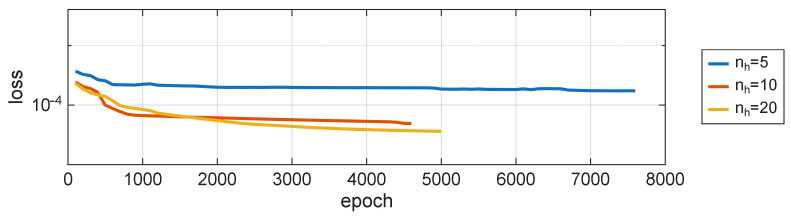
Training loss for three different NARX architectures: $n_{a}=n_{b}=4$ and $n_{h}=5,10,20$.

**Figure 9 sensors-25-07193-f009:**
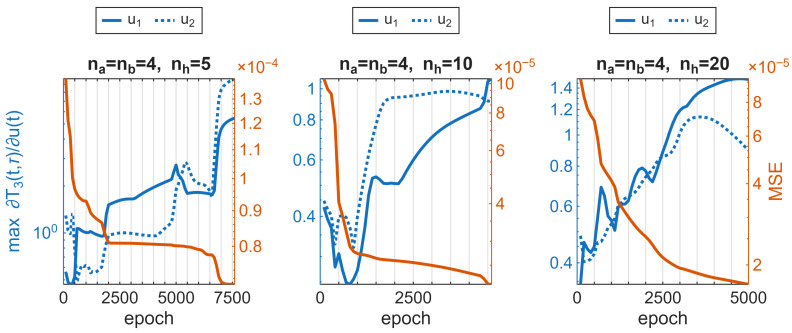
(Color online) Comparison of MSE and maximum derivatives $\partial T_{3}\left(t,\tau \right)/\partial u(t)$ with respect to time and delay τ, in relation to training epoch, plotted for models with different numbers of hidden neurons $n_{h}$.

**Figure 10 sensors-25-07193-f010:**
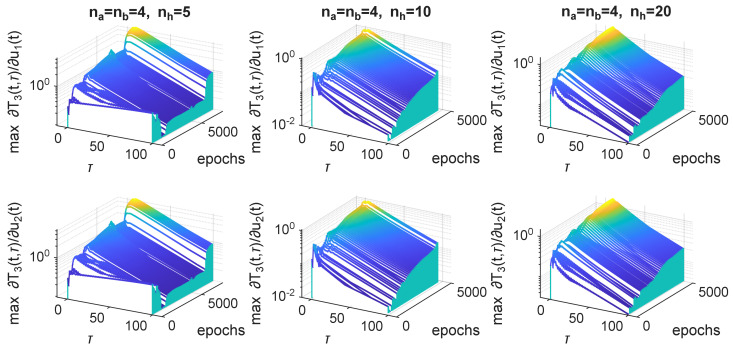
Maximum derivatives $\partial T_{3}\left(t,\tau \right)/\partial u_{1}\left(t\right)$ and $\partial T_{3}\left(t,\tau \right)/\partial u_{2}\left(t\right)$ with respect to time, in relation to τ and training epoch, plotted for models with different numbers of hidden neurons $n_{h}$.

**Figure 11 sensors-25-07193-f011:**
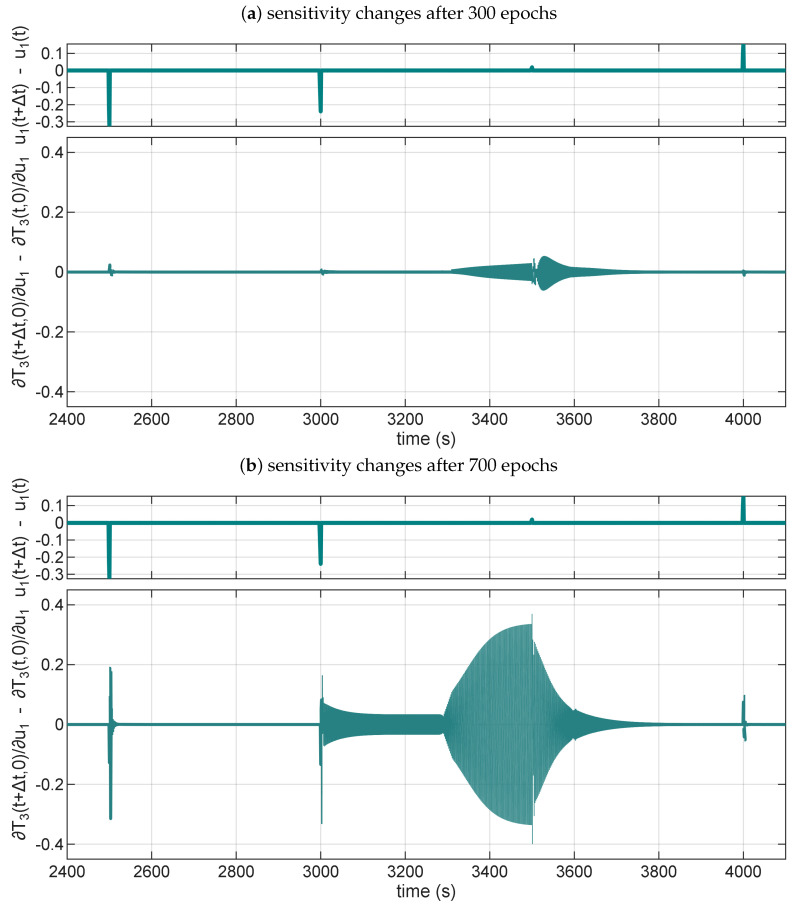
Four states of a NARX model ($n_{a}=n_{b}=4$, $n_{h}=5$) during training: (**a**) a low-sensitivity, undertrained model with low robustness, (**b**) a model with low robustness, (**c**,**d**) robust models with different sensitivities.

**Table 1 sensors-25-07193-t001:** Summary of reviewed papers and their characteristics (key performance indicators related to robustness are written with bold font).

Paper (Year) & Reference	Model Type	Application Domain	Key Performance Indicators (KPIs)
Khan et al., 2015 [30]	Data-driven	Electrochemical sensors	MAE, NMAE, fault prediction, redundancy evaluation
Khan et al., 2016 [31]	Hybrid	Fatigue analysis, predictive maintenance	Prediction error (MAE, RMSE), model comparison
Shao et al., 2019 [32]	Data-driven	Hydrogen production process	RMSE, real-time estimation, time-delay compensation
Guzmán et al., 2019 [13]	Model-based	Greenhouse monitoring	Comparison with a real temperature sensor
Wei et al., 2019 [33]	Data-driven	Semiconductor manufacturing, nonlinear processes	Estimation accuracy, model stability, computational efficiency
Hu et al., 2019 [34]	Data-driven	Building energy fault detection	Classification F1 score, feature reduction, deployment cost
Masti et al., 2021 [19]	Data-driven	Parameter-varying systems, batteries	Estimation accuracy, computational/memory efficiency
Cheng et al., 2021 [14]	Model-based	Greenhouse monitoring	Absolute error between measurement and simulation
Peniak et al., 2022 [35]	Model-based	Safety-related control, industrial IoT	Fault detection, availability, validation accuracy
Heindel et al., 2022 [36]	Data-driven	Semiconductor virtual metrology	Prediction accuracy, multivariate output performance
Hui et al., 2022 [37]	Model-based	Vibro-acoustic monitoring	Estimation accuracy, observability, sensor placement
Li et al., 2023 [38]	Data-driven	Industrial process quality monitoring	Generalization reliability, accuracy, time-delay handling
Stavropoulos et al., 2023 [39]	Hybrid	Drilling, real-time modeling	Prediction accuracy, retraining efficiency
Yao et al., 2024 [40]	Data-driven	IoT, smart home (temperature/humidity)	Model accuracy, error metrics, computational cost
González-Herbón et al., 2024 [1]	Data-driven	Hydraulic thermal plant	Integral of the absolute error, integrated absolute variation
Li et al., 2024 [23]	Data-driven	Waste-water treatment	Root mean squared error, mean absolute error
Hossain et al., 2024 [24]	Data-driven	Pressurized water reactor	Relative L2 error
Li et al., 2024 [41]	Data-driven	ADAS sensor simulation, automotive	**Model generalization**, prediction error, simulation realism
Barabás et al., 2025 [42]	Data-driven	Autonomous vehicles, ADAS, vehicle dynamics	Functional performance, accuracy, robustness, real-time capability

## Data Availability

The raw data supporting the conclusions of this article will be made available by the authors on request.

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
