# Peer review of "Robustness Analysis of a Fast Virtual Temperature Sensor Using a Recurrent Neural Network Model Sensitivity"

_sensors, 2025, doi:10.3390/s25237193_

Round 1
Reviewer 1 Report
Comments and Suggestions for Authors
The manuscript presents a method for assessing the robustness of a temperature soft sensor based on estimating its sensitivity to the input variables. It does have relevance, but the manuscript has several issues that indicate its rejection at the current version, as follows:
- the literature overview is quite poor and should indicate the gaps that the present work is tackling. In the same section, the authors mention ANNs as the only type of virtual sensor used, so it shall be rewritten;
- lines 65-67 are quite confusing;
- lines 86-88, which should indicate the gap, are very confusing;
- the experimental setup needs a schematic diagram should the various elements used in the modeling. As it is currently described only verbally, the reader has difficulty in visualizing the physical setup;
- lines 182-184 are very confusing;
- in line 223, the authors mention that they are presenting the case na=nb=4 because it is the LEAST INFORMATIVE. This does not seem to make sense;
- in line 230, the statement "it is clearly visible" is overreaching, as it is not really easy to see what the authors claim;
Aside from the above items, the overall goal of the manuscript is not clear. I understood that the authors trained a NARX neural network as a soft sensor that receives as input the signals MOV-12 and MOV-22 from 2 control valves, but it is not clear what these signals represent. Are they the percentual of each valve opening? And why MOV-32 is not used? The output is the temperature T3 and the NN is trained on FEM data. So far, it would be a classical soft sensor application, but the manuscript lacks performance metrics for the trained network. Then, the manuscript proceeds to calculate the derivatives of T3 in relation to the 2 inputs, based on disturbing the inputs and tracking the derivatives, but the relevance of this is not clear. Why a large sensitivity is bad? Finally, the results presented in Figures 4 and 5 are very difficult, if not impossible, to understand.
Smaller issues include:
- The highlights should not be included in the body of the manuscript. Also, they are not very insighful;
- throughout the manuscript, units shall be separated from values;
- in line 34, the latin terrm "i.e." makes no sense. It should be "e.g.";
The English grammar is fine, but the overall style is confusing at some points. The manuscript would greatly benefit from a thorough revision by a native English speaker or by some grammar software.
Author Response
Thank you for the insightful comments. Our responses to remarks are in the attached PDF.

Reviewer 2 Report
Comments and Suggestions for Authors
it can be seen in the docx

Author Response

(The authors gave the same response as above.)

Reviewer 3 Report
Comments and Suggestions for Authors
The manuscript presents a robustness analysis of virtual temperature sensors based on Recurrent Neural Networks (RNNs), specifically a Nonlinear AutoRegressive Exogenous (NARX) model. The authors study the model’s sensitivity with respect to input signals, using automatic differentiation to assess how training epochs, network complexity, and model architecture influence robustness.
The topic is timely and relevant for virtual sensing, industrial monitoring, and control applications, and the paper is well-structured and methodologically solid. However, it would benefit from stronger contextualization, clearer statistical validation, and tighter presentation of results.
However, some major changes need to be considered for publication:
-
The review is too descriptive; it summarizes previous works without critically comparing them. Please, include a comparative analysis table summarizing model types (model-based vs. data-driven), application domains, and key performance indicators.
-
Additionally, recent works on explainable AI in virtual sensing could strengthen the contextual grounding. For example: DOI: 10.1016/j.orp.2024.100308
-
The description of the ventilation system and thermal model is clear, but there’s no quantitative validation of the simulation (e.g., comparison with measured data or uncertainty quantification). Please, include numerical error metrics or reference simulations to demonstrate that the FVM model is accurate enough to generate reliable training data.
-
The choice of the NARX architecture (vs. LSTM or GRU) is not fully justified. Authors mention they will analyze this in future work, but a short rationale here (e.g., computational simplicity, interpretability) would strengthen the section.
-
The training setup could include more quantitative details: learning rate, stopping criteria, and computational resources.
-
The analysis is interesting but lacks statistical depth. The authors show qualitative trends (in Figures 4–8) but do not quantify sensitivity differences (e.g., mean ± std, confidence intervals). Please, add a numerical summary table of peak sensitivities and corresponding epochs for each model configuration.
-
The seeding interval of every 100th sample is arbitrary; briefly explain why this choice was made.
Author Response

(The authors gave the same response as above.)

Round 2
Reviewer 1 Report
Comments and Suggestions for Authors
The authors have properly addressed all the issues indicated in the previous revision, and the manuscript has been much improved. I consider that it is now OK for publication.
Comments on the Quality of English LanguageThe English language is OK now.
Reviewer 3 Report
Comments and Suggestions for Authors
The authors have modified the article according to the reviewer´s comments. For this, the article has been improved in terms of scientific soundness and quality of presentation.